# Pollution Effect of the Agglomeration of Thermal Power and Other Air Pollution-Intensive Industries in China

**DOI:** 10.3390/ijerph20021111

**Published:** 2023-01-08

**Authors:** Jingkun Zhou, Juan Tian, Diandian Zhang

**Affiliations:** 1Business School, Ludong University, 186 Hongqizhong Road, Zhifu District, Yantai 264025, China; 2School of Business Administration, Hebei University of Economics and Business, Shijiazhuang 050061, China; 3School of Public Administration and Policy, Shandong University of Finance and Economics, 7366 Erhuandong Road, Lixia District, Jinan 250014, China

**Keywords:** air pollution-intensive industry, location entropy index, spatial spillover effect, spatial panel Durbin model

## Abstract

With the rapid development of the Chinese economy, air pollution is becoming increasingly serious, which greatly impacts the lives and activities of people, and the spatial agglomeration of thermal power and other air pollution-intensive industries (TPAPIs) is an important factor. The purpose of this article is to find the air pollution-intensive industries scientifically, to analyze the effects of pollution from TPAPIs in China, and to provide a basis for the planning and adjustment of TPAPIs. In this study, an air pollution index was adopted to identify TPAPIs, a location quotient was employed to measure the agglomeration of TPAPIs in China, and the global Moran’s I index was determined to examine the spatial agglomeration characteristics of these industries and the spatial characteristics of air pollution. On this basis, a spatial panel Durbin model describing atmospheric pollution was constructed. The pollution effects of the agglomeration of TPAPIs were examined in regard to spatial agglomeration and spillover effects. In the study, it was found that the agglomeration of TPAPIs in different regions of China exhibited a significant positive spatial correlation, and spatial dependence becomes increasingly notable. A significant inverted U-shaped relationship was found to exist between the spatial agglomeration of TPAPIs and air pollution, and thus the spatial agglomeration of TPAPIs imposes a significant spatial spillover effect on air pollution. Specific policy suggestions are proposed, such as the formulation of science-based policies targeting TPAPIs, promotion of interregional cooperation, establishment of a regional joint prevention and control mechanism, and effective elimination of the excess capacity of outdated TPAPIs.

## 1. Introduction

Due to the COVID-19 pandemic starting on 23 January 2020, people stayed home as a precaution, and their motor vehicles remained parked in neighborhoods. Factories and construction sites remained idle; coal, as a replacement of gas, was the main utilized energy source, fireworks were forbidden, and restaurants were shut down. Despite these measures, most of the cities in northern China suffered two haze pollution periods in 2020, from 25 to 28 January and from 11 to 13 February. This mainly occurred because of the heating demand in winter and because steel, thermal power, petrochemical, and other enterprises continued their operations. Moreover, the climate did not allow effective pollutant dispersal, and a spillover effect occurred due to the excessive agglomeration of steel, thermal power, petrochemical, and other industries. As a result, it is of great importance to determine an appropriate way to agglomerate such industries causing air pollution.

Industrial agglomeration is an important method to improve the regional economy because it yields an agglomeration effect that improves the utilization of resources and optimizes their allocation. However, agglomeration may also produce a large number of pollutants. With the coordinated development of the economy and environment, increasing research has been performed on the relationship between industrial agglomeration and environmental pollution, and it can be divided into four categories according to its different viewpoints. The first viewpoint is that agglomeration aggravates pollution because it causes massive emissions of pollutants [1,2]; the denser factories are, the more serious the resulting pollution [3,4]. The second viewpoint is that the spillover of knowledge and technology endows the agglomeration process with positive externalities [5]. The third viewpoint is that there exists an inverted U-shaped relationship between industrial agglomeration and pollution [6,7]. Finally, the fourth viewpoint is that agglomeration generates a significant spatial spillover effect on pollution [8], and economic agglomeration imposes the same effect [9].

In summary, scholars in China have researched the influences of industrial agglomeration on environmental pollution, and they have considered the above four viewpoints, which are presented as a reference. However, there are not enough studies on the spatial effect of industrial agglomeration, especially research on the spatial spillover effect of the agglomeration of TPAPIs on environmental pollution. TPAPIs, which directly cause air pollution, were selected as objects of study. Based on the identification of these industries and the influences of spatial factors on air pollution, spatial econometrics was applied as a method to examine the environmental effect of the agglomeration of TPAPIs in each province of China from 2000 to 2016 from the perspective of air pollution spillover. This study aims to facilitate the spatial planning and adjustment of TPAPIs in China. Further material is divided into several sections. Section 2 focuses on the research hypotheses, Section 3 focuses on the research methodology, Section 4 contains an examination of spatial correlation and the setup of the spatial model, Section 5 focuses on the research analysis of the article, and Section 6 points out conclusions and policy suggestions.

## 2. Research Hypotheses

**Hypothesis** **1.**
*There is an inverted U-shaped relationship between the agglomeration of TPAPIs and air pollution.*


Industrial agglomeration is conducive to cost savings and utilizes industrial advantages, which is very important for promoting regional economic development. Industrial agglomeration can bring a scaling effect to pollution management, which is conducive to improving the rate of waste utilization and comprehensive pollution treatment [10]. Jalil and Feridun [11] found that industrial agglomeration can promote the emission of pollutants, but with the increase of agglomeration level, the emission of pollutants is significantly reduced. Zhang Ke and Wang Dongfang [12] showed that industrial agglomeration and pollution have a relationship of promotion and then inhibition. Wu Chuanqing et al. [13] verified the nonlinear relationship between manufacturing agglomeration and environmental pollution in the Yangtze River Economic Zone. Shen Yue and Ren Yixin [14] found that there are positive and negative externalities of industrial agglomeration. When the negative externality is greater than the positive externality, the crowding effect of industry will be greater than the agglomeration or scale effect, thus aggravating the environmental pollution situation; when the negative externality is less than the positive externality, the agglomeration or scale effect is amplified, and various elements will keep pouring into the agglomeration within achieving the effect of improving environmental pollution. It can be seen that there is an inverted “U” non-linear relationship between the agglomeration of TPAPIs and air pollution. Previous literature has mostly studied the spatial spillover relationship of industrial agglomeration, whereas focusing on the agglomeration of TPAPIs can better identify its effects on pollution.

**Hypothesis** **2.**
*There is a pollution spatial spillover effect from the agglomeration of TPAPIs.*


With industrial agglomeration, the spatial relationship of environmental pollution between regions becomes strong. Hossein and Kaneko [15] showed that there is spatial spillover of environmental pollution between countries and that there is also spatial spillover of environmental pollution in China [16]. They showed that the impact of energy consumption on air pollution has significant local and spatial spillover effects [17]. Further studies have pointed out that collaborative agglomeration has a significant spatial spillover effect in which carbon intensity has a significant spatial convergence effect [18]. It can be seen that there is a pollution spatial spillover effect of thermal power and other air pollution intensive industrial agglomeration, and it is difficult to dig deeper into the pollution effect generated by TPAPI agglomeration while ignoring this spatial influence.

**Hypothesis** **3.**
*There is a significant positive spatial correlation from the agglomeration of TPAPIs.*


Based on the spatial characteristics of pollution-intensive industrial agglomeration and environmental pollution, Liu et al. [19] found that both pollution-intensive industrial agglomeration and environmental pollution have a significant positive spatial correlation. Zhong Juan and Wei Yanjie [20] showed that industrial agglomeration can promote pollution reduction and that the positive spatial correlation is very obvious. Zhang et al. [21] showed that industrial agglomeration further enhances a strong positive spatial spillover effect on haze pollution in local and neighboring areas. Therefore, there is also a significant positive spatial correlation from the agglomeration of TPAPIs.

**Hypothesis** **4.**
*The agglomeration of TPAPIs can bring economic growth, causing the transfer out of population and non-TPAPIs.*


Industrial agglomeration mainly aims to adjust industrial structure, optimize the spatial pattern of industries, and rectify high-energy industries with the ultimate goal of improving the environment and achieving coordinated economic and environmental development. Porter [22] found that industrial agglomeration can optimize industrial institutions and promote the generation and development of environmental industry clusters. Karkalakos [23] argued that industrial agglomeration promotes technological progress and that the technological spillover effect of agglomeration enhances the diffusion of clean technologies [24]. Chen Jianjun and Hu Chenguang [25] found that industrial agglomeration in China also has the effect of optimizing industrial structure. This shows that the agglomeration of TPAPIs can lead to economic growth. The agglomeration and transfer of industries will inevitably lead to the regrouping and mobility of labor. The trajectories of the centers of gravity of the manufacturing industry and of population migration show a “deviation-convergence”, with the centers of gravity of both moving in opposite directions before 2010 and gradually converging after 2010 [26]. The transferring position of high-pollution industries will break the existing industrial pattern while also taking the place of incoming industries. The concentration of pollution caused by the agglomeration of polluting industries will inevitably force other non-polluting industries to transfer out. The agglomeration of TPAPIs results in a higher possibility that non-TPAPIs will transfer out.

## 3. Research Methodology

### 3.1. Identification of TPAPIs

Pollution-intensive industries produce very large quantities of pollutants and cause deterioration in the regional environment during production [27]. TPAPIs produce numerous air pollutants either directly or indirectly during production and processing. At present, there are three main ways to categorize pollution-intensive industries. The first method was proposed by Tobey [28], who categorized these industries according to the ratio of the disposal costs of pollution control to the total cost of production. Becker [29] established the second method, in which these industries were categorized according to their pollutant emission scale. The third method was developed by Mani and Wheeler [30], who categorized these industries according to the number of released pollutants per unit of output. These three methods are not comprehensive because they only consider the disposal cost, scale, and emission intensity, which are all considered in this study. With the *Industrial Classification of National Economic Activities* (GB/T 4754–2017) as a reference, pollution-related indexes of the exhaust gas originating from various industries are determined to identify TPAPIs. The equation of the pollution index of TPAPIs is as follows:(1)$$I_{i}=Q_{i}+G_{i}=\alpha \cdot \frac{E_{i}}{B_{i}}+(1-\alpha )\cdot \frac{E_{i}}{D}$$
where *I_i_* is the air pollution index of industry *i*, *Q_i_* is the emission intensity of industry *i*, *G_i_* is the emission scale of industry *i*, *E_i_* is the number of released pollutants by industry *i*, *B_i_* is the gross output value of industry *i*, *D* is the total quantity of released pollutants, and α equals 0.5 [27]. Based on statistics from 2009, 2011, 2013, and 2015, the air pollution index values of different industries in China were determined. Among them, the six industries with relatively high index values were power and/or heat production and supply, smelting and pressing of ferrous metals, petroleum refining and coking, manufacture of raw chemical materials and products, manufacture of nonmetallic mineral products, and smelting and pressing of nonferrous metals (the data were retrieved from the *China Statistical Yearbook on Environment*). Since 2015, the use of electricity instead of coal has been greatly advocated in the process of nonferrous metal smelting, and many cement plants have been closed to enable their technical transformation. As a result, these two industries have witnessed a very fast reduction in pollutant emissions, especially in areas suffering from severe air pollution. Considering the above aspects, in this study, the remaining four industries are investigated as TPAPIs, including power and/or heat production and supply, smelting of ferrous metals, manufacture of raw chemical materials, and petroleum refining and coking.

### 3.2. Agglomeration Features of TPAPIs

#### 3.2.1. Measurement Methods for the Agglomeration of TPAPIs

The current industrial agglomeration level measurement mainly uses a location entropy index, industry concentration, Herfindahl–Hirschman Index, spatial Gini coefficient, and E–G index. Guan et al. [31] reviewed the above methods and concluded that the location entropy index is the best choice in measuring the specialization advantage of an industry in a region compared with the overall level. In particular, when measuring the level of agglomeration of pollution-intensive industries in multiple provinces and regions across the country, it is more applicable to use the location entropy index method not only to eliminate the scale differences between provinces and regions, but also to reflect the specialization advantages that each province and region has at the national level [19]. Therefore, this study uses location entropy index to measure the agglomeration level of TPAPIs in combination with the characteristics of the research object. The equation applied in this study to compute the location entropy index is as follows:(2)$$loc\_inds_{it}=\frac{m_{it}/M_{it}}{\sum_{i=1}^{30}m_{it}/\sum_{i=1}^{30}M_{it}}$$

In Equation (2), *loc_inds_it_* is the location entropy of the gross value of the industrial output in province *i* over *t* years, *m_it_* is the gross value of the industrial output of the four pollution-intensive industries in province *i* over t years, and *M_it_* is the gross output value of all of the industries in province *i* over *t* years.

#### 3.2.2. Analysis of the Spatiotemporal Evolution of the Agglomeration of TPAPIs

Location entropy can measure the spatial distribution pattern of regional factors to compare the differences in the degree of industrial agglomeration among regions. According to Equation (2), a spatial distribution map of the location entropy of TPAPIs from 2000 to 2016 was generated (Figure 1). The figure shows that location entropy exhibits a downward trend with notable fluctuations.

In terms of high values of location entropy, in 2000, high values were mainly distributed in the Inner Mongolia Autonomous Region, Jilin Province, and Liaoning Province, extending southward to Guizhou Province, the central area of the Guangxi Zhuang Autonomous Region, and the western region, excluding Yunnan Province. This distribution occurred due to the abundant natural resources in these regions, such as coal in the Inner Mongolia Autonomous Region, coal, petroleum, and iron ore in Liaoning Province and Jilin Province, and mineral resources in Guizhou Province and Guangxi Province. Additionally, in 2000, the transportation network in China had not yet been completed, and transportation costs were high, thus TPAPIs were inclined to agglomerate at certain locations. The distribution of the regions with a location entropy index value higher than 1.1 from 2000 to 2006 moved from central and western China to the Beijing–Tianjin–Hebei region. In 2006, high index values only occurred in Hebei Province, which resulted from several factors. In 2000, the Tenth Five-Year Plan proposed a change in the structure of incremental industries and active development of emerging and high-tech industries, which reduced the gross output value of TPAPIs in each province. Hebei enjoys a special location near Beijing and Tianjin with abundant coal and iron ore resources and convenient transportation conditions. Hebei Province strove to develop air pollution-intensive thermal power and steel industries to guarantee the supply of power and steel, respectively, to Beijing and Tianjin, and as a result, the level of observed agglomeration of these industries was relatively high. However, regions with a location entropy index value higher than 1.1 became increasingly dispersed after 2006, which generated a relatively dispersed spatial distribution pattern in 2016. In 2016, high values were observed in various regions, including Hebei Province, Henan Province, Guangxi Zhuang Autonomous Region, Gansu Province, and the Ningxia Hui Autonomous Region. Faced with increasing air pollution in China, people have become more aware of the importance of air quality protection, and each region has proposed strict environmental regulations. Particularly, in the developed regions in central and eastern China, pollution-intensive industries faced high environmental costs. As a result, they moved to provinces such as the Guangxi Zhuang Autonomous Region, Gansu Province, and Ningxia Hui Autonomous Region, which have liberal environmental regulations or a high self-purification ability of the air. Hebei Province and Henan Province contain abundant coal, a less developed economy, and convenient transportation conditions and are located near regions with rapid economic development, thus TPAPIs have agglomerated in these two provinces.

In terms of low location entropy index values, in 2000, low values were largely distributed in the Xinjiang Uygur Autonomous Region, Heilongjiang Province, and coastal provinces in southeastern China, including Fujian Province, Zhejiang Province, and Guangdong Province. This distribution depended on several factors. Although the Xinjiang Uygur Autonomous Region enjoys ample natural resources and liberal environmental regulations, its transportation conditions are inconvenient, resulting in high costs that are to the disadvantage of the agglomeration of pollution-intensive industries. Heilongjiang Province is the northernmost province among the three northeastern provinces in China and suffers from the worst climate and environmental conditions, which indicates that mechanical equipment is difficult to operate, and that production is easily influenced by weather conditions. In terms of the coastal provinces in southeastern China, including Fujian Province, Zhejiang Province, and Guangdong Province, in 2000, these provinces mainly hosted labor-intensive light industries from overseas instead of pollution-intensive industries dominated by heavy industries. After 2000, the distribution of low values was concentrated instead of scattered in border areas such as northeastern, northwestern and southeastern China. By 2016, regions with a location entropy index value lower than 0.9 widely occurred in three northeastern provinces in China, the Inner Mongolia Autonomous Region, Shanxi Province, Gansu Province, and the Xinjiang Uygur Autonomous Region, and in coastal provinces in southeastern China, including Guangdong Province and Zhejiang Province. This distribution was attributed to various factors. Faced with increasingly severe air pollution, China enhanced its regulations on environmental protection. Especially after its economy entered a period of the new normal, China witnessed the transformation of traditional industries, industrial upgrades, and cultivation of emerging industries in various regions, including southeastern coastal provinces, such as Guangdong Province and Zhejiang Province, and central and western regions. Additionally, with increased efforts towards environmental protection, the costs of certain factors in the central and western regions continued to increase, which constrained the activities of pollution-intensive industries in the developed regions in eastern China.

## 4. Examination of Spatial Correlation and Setup of the Spatial Model

### 4.1. Examination of Spatial Correlation

As an obvious spatial spillover effect of the agglomeration of TPAPIs, spatial measures were introduced to examine the local impact of geographically adjacent areas. The examination of spatial correlations is a prerequisite for the setup of spatial econometric models, including global spatial correlation and local spatial correlation. The global spatial correlation examination focuses on describing the correlation of the study area as a whole, and this study is more suitable to adopt the global spatial correlation examination for testing 30 provinces and cities across China. The global spatial correlation examination is usually measured using Moran’s I index.

Air pollution is influenced by various factors, and there exists a certain correlation among them. Global Moran’s I is applied to analyze the global spatial autocorrelation of the location entropy of TPAPIs. The equation of global Moran’s I is expressed as follows:(3)$$I=\frac{n\sum_{i=1}^{n}\sum_{j=1}^{n}w_{ij}\left(x_{i}-\bar{x}\right)\left(x_{j}-\bar{x}\right)}{\sum_{i=1}^{n}\sum_{j=1}^{n}w_{ij}\sum_{i=1}^{n}{\left(x_{i}-\bar{x}\right)}^{2}}=\frac{\sum_{i=1}^{n}\sum_{j\neq 1}^{n}w_{ij}\left(x_{i}-\bar{x}\right)\left(x_{j}-\bar{x}\right)}{S^{2}\sum_{i=1}^{n}\sum_{j\neq i}^{n}w_{ij}}$$

In the above equation, $w_{ij}$ is the spatial proximity of i to j. When *i* and *j* are adjacent, $w_{ij}$ = 1, and conversely, $w_{ij}$ = 0. The value range of global Moran’s *I* is [−1, 1]. The normalized Z statistic is adopted to examine whether autocorrelation occurs between object regions. If Z is greater than zero, then there exists a positive autocorrelation between objects, and the higher Z is, the higher the existing positive correlation. If Z is less than zero, then there exists a negative autocorrelation between objects, and the lower Z is, the higher the existing negative correlation. If Z equals zero, then this indicates that the observed values are independently and randomly distributed.

According to the availability of data, the emissions of the polluting industry can also be determined. The emissions of SO_2_ and industrial SO_2_ are considered to be measures of air pollution, and global Moran’s I was adopted to examine the spatial autocorrelation between air pollution and the agglomeration of TPAPIs. The results are listed in Table 1.

In terms of the Moran index of SO_2_, the *p*-values were all less than 0.01, whereas the Moran index of SO_2_ increased from 0.213 in 2000 to 0.506 in 2016, indicating that the spatial spillover effect of SO_2_ is more obvious.

Looking at the Moran index of industrial SO_2_, its conclusion was similar to that of SO_2_, indicating that the spatial spillover effect of industrial SO_2_ is also more obvious. Meanwhile, the spatial spillover effect of *loc_inds* also has similar conclusions.

According to the temporal evolution features of global Moran’s I, as indicated in Table 1 above, Moran’s I of the location entropy of SO_2_ emissions, industrial SO_2_ emissions, air pollution, and TPAPIs continues to increase, which indicates the gradual increase in SO_2_ emissions, industrial SO_2_ emissions, and level of spatial agglomeration of TPAPIs. These results may be closely related to the spatial agglomeration of economic growth. Driven by economy of scale and learning effects, enterprises with similar products or related enterprises are inclined to be adjacent to each other, which causes a high spatial agglomeration of TPAPIs and air pollution. This indicates that there is a significant positive spatial correlation among air pollution-intensive industries such as thermal power, which is consistent with Hypothesis 3. It is evident that in regards to the spatial agglomeration of air pollution, other factors should also considered, including the spatial heterogeneity of the resource distribution, the increasingly important transportation location, and population migration to cities.

### 4.2. Setup of the Spatial Model

Considering that the spatial panel error model (SEM) and spatial panel lagged model (SLM) were to be incorporated into one model simultaneously in this study, the spatial panel Durbin model (SDM) was used. Because the coefficient estimates of the explanatory variables in the spatial regression model did not represent the true partial regression coefficients, it had to be decomposed into direct and indirect effects using partial differentiation [32]. In this study, direct effects were used for the average effect of all regional explanatory variables on the explanatory variables in the region, and indirect effects were used for the average effect of all regional explanatory variables on the explanatory variables in neighboring regions.

The spatial autoregressive model set up in this study is as follows:(4)pullit=α+ρWpullit+β1loc_indsit+β2loc_indit^2+X∗λ+μi+νt+ξit
where i denotes the region, t is the time (year), and pull is the index of air pollution, which is measured according to the natural logarithm of the total emissions of SO_2_ and industrial SO_2_ in this study. Moreover, *loc_inds* is the agglomeration of TPAPIs, *loc_inds*^2 is the square of the agglomeration, and X is the matrix of other controlled variables excluding the agglomeration of air pollution. In addition, μ is the regional fixed effect, which is applied to control the change in several factors with the region, ν is the time fixed effect, which is applied to control the change in several factors over time, ξ is the random disturbance term, and W is the spatial weight matrix. The parameters considered in this study include *ρ*, *β*_1_, and *β*_2_. If *ρ* is greater than zero, then this indicates that there exists a spatial spillover effect in regard to air pollution. If *β*_1_ is greater than zero and *β*_2_ is less than zero, then this indicates that there exists an inverted U-shaped relationship between air pollution and agglomeration. Moreover, based on the spatial autoregressive model, the spatial Durbin model was applied to research the relation between TPAPIs and air pollution. The model is expressed as follows:(5)pullit=α+ρWpullit+β1loc_indsit+β2loc_indsit^2+X∗λ+β3Wloc_indsit+β4Wloc_indsit^2+WX∗γ+μi+νt+ξit

The parameters in Equation (5) are the same as those in Equation (4).

### 4.3. Selection of the Variables and Explanation

To avoid possible regression errors caused by the omission of variables, five controlled variables are introduced into the model, including the population density, energy efficiency, level of economic development, proportion of fiscal revenue to the expenditure, and foreign direct investment, which are explained in detail below.

Level of economic development (lnpgdp). The natural logarithm of the gross domestic product (GDP) per capita was applied to measure the real GDP per capita of each province. There exists an inverted U-shaped relationship between the levels of economic development and environmental pollution [33,34]. Scholars have also determined that an N-shaped relation occurs between the comprehensive pollution index and GDP per capita, which indicates that with increasing economic growth, the environmental quality first deteriorates, then improves, and then finally worsens [35].

Foreign direct investment (fdi_gdp). The proportion of real foreign direct investment in the GDP was measured. There are two hypotheses regarding the relation between foreign direct investment and environmental pollution: one is the pollution haven hypothesis [36], and the other is the pollution halo hypothesis [37].

Population density (pop_den). The number of people per unit land area was measured to represent the population density. There are three types of findings regarding population density and environmental pollution. The first finding is that there exists a positive correlation between these aspects [38], the second finding is that a negative correlation occurs [39], and the third finding is that there exists an N-shaped relationship [40].

Energy efficiency (elecity_gdp). The electricity consumption per unit GDP was measured to represent energy efficiency. If energy efficiency is promoted, then ecosystem deterioration is relieved and energy consumption is reduced [41]. Environmental pollution is alleviated if the energy consumption structure is optimized and energy efficiency is enhanced [42].

Proportion of fiscal revenue to expenditure (fin_inc_exp). The quotient of fiscal revenue divided by fiscal expenditure was determined. In past studies, scholars have proposed that an increase in fiscal revenue aggravates environmental pollution [43], whereas others have reported that there exists a positive relation between fiscal expenditure and environmental pollution [44].

The controlled variables in this study were extracted from the *China Statistical Yearbook*, *China Statistical Yearbook on Environment* and EPS Database. In addition, to avoid personal statistical errors and control the heteroscedasticity, the natural logarithms of the controlled variables were applied. The descriptive results of each variable are listed in Table 2 below.

## 5. Analysis of the Results

### 5.1. Analysis of the Results of the Spatial Autoregressive Model

An analysis of the results of the spatial autoregressive model in this study is summarized in Table 3 below.

The spatial autoregressive model analysis results (Table 3) reflect the significant statistical tests of the estimated values of the spatial lag parameters of $lnSO_{2}$ and $lnindSO_{2}$, which demonstrate that there exists an evident spatial correlation between SO_2_ and industrial SO_2_ emissions in the provinces, municipalities and autonomous regions (excluding the Tibet Autonomous Region due to the loss of data), 30 regions in total, of mainland China. In particular, the correlation indicates that the emissions of SO_2_ and industrial SO_2_ in a given province or municipality are influenced by those in its adjacent provinces or municipalities. As in Hypothesis 2, there is also a significant spatial spillover effect from TPAPIs.

According to the results, it is evident that *loc_inds* is positively correlated with $lnSO_{2}$ and $lnindSO_{2}$, whereas *loc_inds^2* is negatively correlated with $lnSO_{2}$ and $lnindSO_{2}$. This verifies that an inverted U-shaped relationship occurs between the sales of TPAPIs and SO_2_ and industrial SO_2_ emissions. This is consistent with the description of Hypothesis 1. These industries emit a large amount of sulfur dioxide during production. If factories are dense, then emissions rapidly increase, which leads to an increase in regional emissions. However, according to the economy of scale effect, when industries are sufficiently concentrated, production becomes increasingly efficient as scale increases. As a result, the overall level of emissions decreases. Additionally, due to the spillover effect, the exchange of technology and knowledge among enterprises also lowers emissions because the advancement of energy consumption reduction and production technology is facilitated. Moreover, this enables enterprises to enhance their technological innovation, improve the efficiency of centralized treatment, and comprehensively recycle pollutants.

*Fdi_gdp* is negatively correlated with $lnSO_{2}$ and $lnindSO_{2}$, which contradicts the pollution haven hypothesis that states that pollution-intensive industries tend to be located in regions with liberal environmental regulations. This point is not supported by the SO_2_ emissions in China mainly because most emissions originate from heavy industries, including power plants, residential coal, steel, and petrochemical plants. Most of these industries are operated under the state-owned economy, and the proportion of foreign direct investment is relatively low. It is evident that pgdp is positively correlated with both $lnSO_{2}$ and $lnindSO_{2}$, especially in the model of $lnindSO_{2}$, which indicates that although TPAPIs emit very large amounts of SO_2_, they remain the main sources of regional economic growth. It is clear that the levels of fiscal revenue and expenditure are positively correlated with both $lnSO_{2}$ and $lnindSO_{2}$, which, however, does not indicate that fiscal revenue significantly impacts SO_2_ emissions. This only reflects the correlation between fiscal revenue and SO_2_ emissions, and it is possible that the increase in emissions tends to represent the dynamism of regional economic activities and improvement in economic development. Population density exhibits a significant positive correlation with $lnSO_{2}$ and no evident correlation with $lnindSO_{2}$. This may occur because with increasing population density, the consumption of coal and other commodities rises, which then increases SO_2_ emissions. It is evident that energy efficiency is positively correlated with both $lnSO_{2}$ and $lnindSO_{2}$, which demonstrates that the higher electricity consumption per unit GDP is, the higher SO_2_ and industrial SO_2_ emissions are.

### 5.2. Analysis of the Results of the Spatial Durbin Model

The estimation and examination results of the spatial Durbin model are listed in Table 4 below.

According to Table 4, the spatial Durbin model yielded similar results to those of the spatial autoregressive model. The following focuses on the analysis of the results of the spatial spillover terms of each explanatory variable. There are significant positive correlations between *loc_inds* and the spatial spillover terms of $lnSO_{2}$ and lnindso2, whereas significant negative correlations occur between *loc_inds^2* and the spatial spillover terms of $lnSO_{2}$ and lnindso2, which indicates that TPAPIs lead to an increase in SO_2_ emissions not only in their regions, but also in surrounding regions. The spatial spillover, in terms of the levels of economic development and population density, reveals significant negative correlations with the SO_2_ emissions in local regions, which mainly occurs because the increase in emissions causes the migration of population and enterprises, leading to a decline in the population density and level of economic development in surrounding regions. This suggests that the emissions from the agglomeration of TPAPIs have an economic and demographic impact, which is consistent with Hypothesis 4. There exists a significant positive correlation between the spillovers in terms of the fiscal revenue and in terms of SO_2_ emissions in a given region. The possible reason is that economic growth leads to local fiscal revenue improvement, and the economic growth in surrounding regions increases SO_2_ emissions, thus resulting in more SO_2_ in local regions due to diffusion.

### 5.3. Analysis of the Effect Decomposition Results of the Spatial Durbin Model

The effect decomposition results of the spatial Durbin model in this study are summarized in Table 5 below.

According to Table 5, in terms of the direct effect, the influence coefficients of the location entropy of TPAPIs in regards to the emissions of SO_2_ and industrial SO_2_ in a given region are 1.647 and 2.078, respectively, according to the significance test at the 1% level, which indicates that spatial agglomeration of TPAPIs is the main cause of the SO_2_ emissions. In terms of their indirect effects, the spatial spillover effects of these industries regarding the emissions of SO_2_ and industrial SO_2_ are 5.706 and 5.355, respectively, through significance testing at the 1% level, which demonstrates that the SO_2_ emissions in a given region are influenced by TPAPIs, both locally and in their surrounding areas. This may occur due to the significant spatial spillover effects of air pollution and diffusion of air pollutants, which is consistent with Hypothesis 2.

In terms of the direct and indirect effects of the controlled variables, foreign direct investment imposes a significant negative direct effect, whereas the level of economic development yields a significant positive direct effect, and both variables exert no evident indirect effects. This indicates that these two variables only produce significant impacts on local SO_2_ emissions. Population density imposes a significant positive direct effect and negative indirect effect, and the absolute value of the latter is much higher than that of the former. This indicates that complex interactions occur between the local emissions of SO_2_ and the population density in surrounding regions, which may be because local SO_2_ emissions, especially residential SO_2_ emissions, affect the perception of the residents in surrounding regions towards their dwelling environment, thus leading to their emigration. Energy efficiency exhibits significant positive correlations with both direct and indirect effects, which indicates that the emissions of SO_2_ mainly originate from energy consumption for economic development in local and surrounding regions.

In terms of the total effect, the location entropy of TPAPIs imposes significant positive effects, which are the second-highest, on SO_2_ emissions, which indicates that these TPAPIs are the main sources of SO_2_, especially industrial SO_2_. Economic development and fiscal revenue and expenditure yield significant positive effects on the emissions of SO_2_, which verifies that local economic growth generates SO_2_, and issues of sacrificing the environmental quality remain in the economic development model of China. Energy efficiency produces significant positive effects, which are the highest, on the emissions of SO_2_, which indicates that the technical level and energy consumption remain obstacles in the development and transformation of the economy. Energy consumption reduction through technological promotion is an important starting point to develop science and establish an environmentally friendly society.

## 6. Conclusions and Policy Suggestions

### 6.1. Conclusions

First, there exists a significant inverted U-shaped relation between the agglomeration of TPAPIs and SO_2_ pollution. High agglomeration increases the emissions of air pollutants, such as SO_2_, in the short term, thus aggravating air pollution. However, when the agglomeration level exceeds a certain level, the emissions of air pollutants are reduced owing to the scale and technological effects of related industries and the promotion of environmental protection.

Second, the agglomeration of TPAPIs leads to a significant spatial spillover effect of SO_2_ pollution. Due to airflow and diffusion of air pollutants, pollutants such as SO_2_ emitted by TPAPIs pollute the air environment in both local and surrounding regions.

Third, the agglomeration of TPAPIs exhibits a significant positive spatial correlation. The emissions of sulfur dioxide and agglomeration of TPAPIs in neighboring provinces tend to be correlated.

Finally, the agglomeration of TPAPIs facilitates increases to GDP and fiscal revenue but also leads to the emigration of residents and non-TPAPIs. Because China is still in an extensive stage of development, agglomeration may cause enhancement of the regional GDP and fiscal revenue. However, these industries may also easily cause environmental pollution because they lack enough technology, consume much energy, and emit many air pollutants. With the improvement of the economy and living standards, people are paying increasing attention to their health. Therefore, residents and non-TPAPIs in affected regions are likely to emigrate due to very high pollutant emissions.

### 6.2. Policy Suggestions

#### 6.2.1. Formulate Plans for the Agglomeration of TPAPIs in an Appropriate Way

It is recommended to analyze the carrying capacity of the regional environment with appropriate planning methods, such as location analysis, strength, weakness, opportunity, and threat (SWOT) analysis, analysis of the status of regional natural resources, and investigation of the status of industrial development, according to features of the environmental self-purification abilities of different regions and the influence mechanisms and effect analyses of different types of TPAPIs, including thermal power, steel, petroleum, and chemistry. It is essential to formulate development plans for these industries in terms of spatial layout according to the self-purification ability of the region and its environment.

#### 6.2.2. Promote Interregional Cooperation and Construct Governance Mechanisms for Joint Defense and Control among Regions

China should promote the construction of governance mechanisms for joint defense and control among regions to avoid leaving them to fend for themselves. First, it is necessary to improve the legislative system and integrate the concept of joint defense and control into enhanced laws and norms in regards to regional prevention and control of air pollution. Second, it is essential that regulations are improved through optimization of management systems, improvement of supervision of regulatory organs, and legislation of regulatory activities. Third, a regional compensation system should be established to avoid the traditional path of pollution before treatment via the comprehensive support of capital, technology, experience, and talent in less developed regions.

#### 6.2.3. Eliminate the Excess Capacity of Outdated TPAPIs in an Efficient Way

It is important to accelerate the elimination of the excess capacity of outdated TPAPIs. First, a management system should be established that regards enterprises as the main body and the market as a guide and relies on the government to promote the appropriate elimination of outdated enterprises with excess capacity. It is also necessary to categorize enterprises with excess capacity according to their intensity of pollution emissions, profitability, production efficiency, and social utility. Moreover, it is recommended to accelerate the shuttering of outdated enterprises with excess capacity in regions with high agglomeration and serious air pollution to revitalize the resources remaining in the outdated excess capacity. Finally, the threshold to enter the market should be raised for TPAPIs, and new projects should be approved after strict examination according to both the layout of related industries and to relevant plans.

## Figures and Tables

**Figure 1 ijerph-20-01111-f001:**
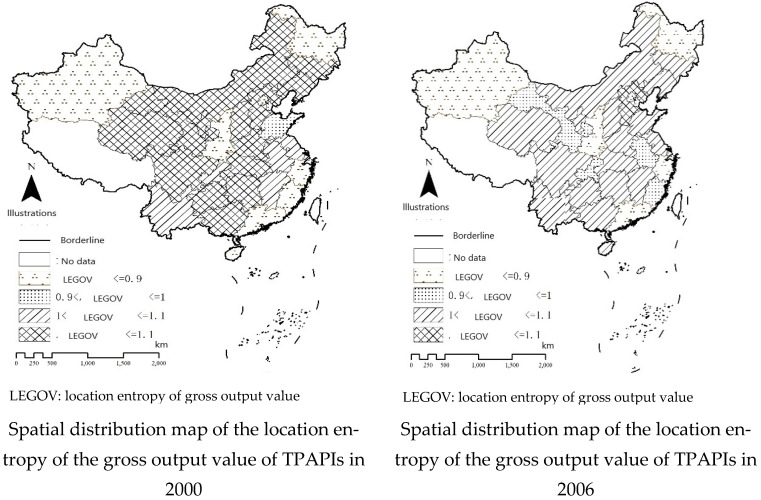
Spatial Distribution of the Location Entropy of the Gross Output of TPAPIs from 2000 to 2016.

**Table 1 ijerph-20-01111-t001:** Moran’s Index of the Location Quotient of the Air Pollution and TPAPIs.

Year	SO_2_	Industrial SO_2_	*loc_inds*	Year	SO_2_	Industrial SO_2_	*loc_inds*
2000	0.213 ***	0.232 ***	0.198 ***	2009	0.378 ***	0.400 ***	0.312 ***
2001	0.223 ***	0.257 ***	0.200 ***	2010	0.396 ***	0.420 ***	0.324 ***
2002	0.246 ***	0.287 ***	0.212 ***	2011	0.420 ***	0.432 ***	0.354 ***
2003	0.272 ***	0.310 ***	0.238 ***	2012	0.427 ***	0.460 ***	0.374 ***
2004	0.274 ***	0.321 ***	0.242 ***	2013	0.435 ***	0.481 ***	0.394 ***
2005	0.276 ***	0.347 ***	0.257 ***	2014	0.443 ***	0.485 ***	0.411 ***
2006	0.306 ***	0.349 ***	0.259 ***	2015	0.474 ***	0.492 ***	0.416 ***
2007	0.329 ***	0.370 ***	0.290 ***	2016	0.506 ***	0.511 ***	0.423 ***
2008	0.353 ***	0.384 ***	0.302 ***				

*** *p* < 0.01.

**Table 2 ijerph-20-01111-t002:** Model Variable Selection and Descriptive Analysis.

Variables		Sample Size	Average Value	Standard Deviation	Minimum	Maximum
Explanatory variables						
$lnSO_{2}$	Natural logarithm of the SO_2_ emissions	510	13.170	0.918	9.741	14.509
$lnindSO_{2}$	Natural logarithm of the industrial SO_2_ emissions	510	13.042	0.953	8.241	14.509
Core explanatory variables						
*loc_inds*	Location entropy of the TPAPIs (sales)	510	1.061	0.339	0.330	2.105
*loc_inds^2*	Square of the location entropy of the TPAPIs (sales)	510	1.241	0.787	0.109	4.432
Controlled variables						
*fdi_gdp*	Foreign direct investment	510	0.438	0.551	0.100	5.846
*lnpgdp*	Level of economic development	510	12.705	1.193	9.842	15.969
*fin_inc_exp*	Proportion of fiscal revenue to the expenditure	510	0.798	0.459	−0.254	3.351
*pop_den*	Population density	510	5.417	1.240	1.946	8.245
*elecity_gdp*	Energy efficiency	510	0.131	0.081	0.040	0.521

**Table 3 ijerph-20-01111-t003:** Results of Spatial Autoregressive Model Analysis.

	(1)	(2)	(3)	(4)
	*lnSO_2_*	*lnSO_2_*	*lnindSO_2_*	*lnindSO_2_*
Main				
*loc_inds*	0.273 ***	0.620 ***	0.344 ***	1.109 ***
	(0.050)	(0.189)	(0.059)	(0.222)
*loc_inds^2*		−0.145 *		−0.318 ***
		(0.076)		(0.089)
*fdi_gdp*	−0.077 ***	−0.079 ***	−0.064 **	−0.070 **
	(0.026)	(0.026)	(0.031)	(0.031)
*lnpgdp*	0.041 *	0.039 *	0.074 ***	0.072 ***
	(0.022)	(0.022)	(0.026)	(0.026)
*fin_inc_exp*	0.204 ***	0.199 ***	0.173 ***	0.164 ***
	(0.036)	(0.036)	(0.042)	(0.042)
*pop_den*	0.329 **	0.329 **	0.177	0.180
	(0.151)	(0.151)	(0.178)	(0.176)
*elecity_gdp*	1.357 ***	1.257 ***	1.065 **	0.846 *
	(0.425)	(0.428)	(0.502)	(0.500)
Spatial				
ρ	0.674 ***	0.669 ***	0.557 ***	0.542 ***
	(0.031)	(0.031)	(0.042)	(0.042)
Variance				
sigma2_e	0.038 ***	0.037 ***	0.052 ***	0.051 ***
	(0.002)	(0.002)	(0.003)	(0.003)
*N*	510	510	510	510
*With_R2*	0.113	0.133	0.173	0.217

The standard errors are noted in parentheses. * *p* < 0.1, ** *p* < 0.05, and *** *p* < 0.01.

**Table 4 ijerph-20-01111-t004:** Results of Spatial Durbin Model Analysis.

	(1)	(2)	(3)	(4)
	*lnSO_2_*	*lnSO_2_*	*lnindSO_2_*	*lnindSO_2_*
Main				
*loc_inds*	0.423 ***	1.115 ***	0.421 ***	1.622 ***
	(0.057)	(0.187)	(0.069)	(0.225)
*loc_inds^2*		−0.295 ***		−0.505 ***
		(0.075)		(0.089)
*fdi_gdp*	−0.094 ***	−0.095 ***	−0.080 ***	−0.083 ***
	(0.025)	(0.024)	(0.030)	(0.029)
*lnpgdp*	0.626 ***	0.665 ***	0.652 ***	0.717 ***
	(0.099)	(0.097)	(0.120)	(0.117)
*fin_inc_exp*	0.149 ***	0.140 ***	0.104 **	0.091 **
	(0.035)	(0.034)	(0.042)	(0.041)
*pop_den*	1.028 ***	1.047 ***	0.795 ***	0.838 ***
	(0.182)	(0.178)	(0.219)	(0.213)
*elecity_gdp*	1.812 ***	2.191 ***	1.187 **	1.528 ***
	(0.411)	(0.416)	(0.494)	(0.497)
Wx				
*loc_inds*	−0.370 ***	1.540 ***	−0.123	2.070 ***
	(0.087)	(0.402)	(0.105)	(0.488)
*loc_inds^2*		−0.762 ***		−0.870 ***
		(0.157)		(0.190)
*fdi_gdp*	0.071	0.074	0.054	0.063
	(0.074)	(0.072)	(0.089)	(0.086)
*lnpgdp*	−0.458 ***	−0.440 ***	−0.497 ***	−0.476 ***
	(0.100)	(0.098)	(0.122)	(0.120)
*fin_inc_exp*	0.370 ***	0.382 ***	0.499 ***	0.516 ***
	(0.087)	(0.085)	(0.103)	(0.101)
*pop_den*	−1.775 ***	−1.812 ***	−0.986 **	−1.028 ***
	(0.341)	(0.332)	(0.410)	(0.397)
*elecity_gdp*	1.092	2.008 **	0.701	2.205 *
	(0.954)	(0.952)	(1.147)	(1.138)
Spatial				
ρ	0.657 ***	0.640 ***	0.549 ***	0.504 ***
	(0.032)	(0.032)	(0.044)	(0.046)
Variance				
sigma2_e	0.033 ***	0.031 ***	0.048 ***	0.045 ***
	(0.002)	(0.002)	(0.003)	(0.003)
N	510	510	510	510
*With_R2*	0.316	0.386	0.281	0.376

The standard errors are noted in parentheses. * *p* < 0.1, ** *p* < 0.05 and *** *p* < 0.01.

**Table 5 ijerph-20-01111-t005:** Effect Decomposition of the Spatial Durbin Model.

*LR_Direct*				
*loc_inds*	0.402 ***	1.647 ***	0.442 ***	2.078 ***
	(0.060)	(0.250)	(0.070)	(0.265)
*loc_inds^2*		−0.519 ***		−0.684 ***
		(0.101)		(0.107)
*fdi_gdp*	−0.094 ***	−0.088 ***	−0.080 **	−0.076 **
	(0.031)	(0.029)	(0.033)	(0.031)
*lnpgdp*	0.625 ***	0.662 ***	0.640 ***	0.699 ***
	(0.090)	(0.089)	(0.108)	(0.107)
*fin_inc_exp*	0.264 ***	0.251 ***	0.202 ***	0.178 ***
	(0.042)	(0.040)	(0.046)	(0.043)
*pop_den*	0.761 ***	0.789 ***	0.704 ***	0.754 ***
	(0.196)	(0.194)	(0.225)	(0.216)
*elecity_gdp*	2.434 ***	2.978 ***	1.489 **	1.981 ***
	(0.564)	(0.573)	(0.609)	(0.610)
*LR_Indirect*				
*loc_inds*	−0.247	5.706 ***	0.216	5.355 ***
	(0.205)	(1.157)	(0.193)	(0.998)
*loc_inds^2*		−2.401 ***		−2.081 ***
		(0.462)		(0.396)
*fdi_gdp*	0.018	0.043	0.015	0.047
	(0.187)	(0.182)	(0.172)	(0.160)
*lnpgdp*	−0.121	−0.040	−0.284 **	−0.215 *
	(0.134)	(0.130)	(0.135)	(0.128)
*fin_inc_exp*	1.253 ***	1.199 ***	1.142 ***	1.049 ***
	(0.213)	(0.210)	(0.199)	(0.186)
*pop_den*	−3.009 ***	−2.909 ***	−1.189	−1.138 *
	(0.816)	(0.806)	(0.751)	(0.684)
*elecity_gdp*	6.414 **	8.613 ***	3.047	5.500 **
	(2.838)	(2.646)	(2.542)	(2.250)
*LR_Total*				
*loc_inds*	0.155	7.354 ***	0.659 ***	7.433 ***
	(0.227)	(1.343)	(0.212)	(1.172)
*loc_inds^2*		−2.920 ***		−2.765 ***
		(0.540)		(0.470)
*fdi_gdp*	−0.076	−0.045	−0.065	−0.028
	(0.208)	(0.201)	(0.190)	(0.175)
*lnpgdp*	0.504 ***	0.622 ***	0.356 ***	0.484 ***
	(0.137)	(0.131)	(0.124)	(0.112)
*fin_inc_exp*	1.517 ***	1.450 ***	1.344 ***	1.227 ***
	(0.240)	(0.236)	(0.225)	(0.208)
*pop_den*	−2.248 **	−2.120 **	−0.485	−0.384
	(0.909)	(0.909)	(0.835)	(0.771)
*elecity_gdp*	8.848 ***	11.592 ***	4.535	7.482 ***
	(3.275)	(3.081)	(2.973)	(2.662)

The standard errors are noted in parentheses. * *p* < 0.1, ** *p* < 0.05 and *** *p* < 0.01.

## Data Availability

Data available in a publicly accessible repository.

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
