# Peer review of "Pollution Effect of the Agglomeration of Thermal Power and Other Air Pollution-Intensive Industries in China"

_ijerph, 2023, doi:10.3390/ijerph20021111_

Round 1

Reviewer 1 Report

Presented article „Pollution Effect of the Agglomeration of Thermal Power etc Thermal Power etc Air Pollution-Intensive Industries in China“ focuses on the identification of air pollution from thermal power plants and from energy-intensive industry based on the air pollution index. Authors created a spatial panel Durbin model describing atmospheric pollution. In the study, it is found that the agglomeration of Thermal Power etc Air Pollution-Intensive Industries in different regions of China exhibits a significant positive spatial correlation, and there exists a significant inverted U-shaped relationship between the spatial agglomeration of Thermal Power etc Air Pollution-Intensive Industries and air pollution, and spatial agglomeration of Thermal Power etc Air Pollution-Intensive Industries imposes a significant spatial spillover effect on air pollution. There are specific policy suggestions such as the formulation of science- based policies targeting Thermal Power etc Air Pollution-Intensive Industries, promotion of inter-regional cooperation, establishment of a regional joint prevention and control mechanism, and effective elimination of the excess capacity of outdated Thermal Power etc Air Pollution-Intensive Industries.

The submitted article does not have significant shortcomings, but it is necessary to correct or add the following:

Line 97 – why does α equals 0.5?

Table 1 – what does thy symbol *** mean in this table?

The presented manuscript focuses on the assessment of the impact of produced emissions from industrially developed areas of China to areas with lower industrialization. The results of the mathematical model are subsequently presented in the conclusions. I consider the absence of source data (e.g. number of pollution sources and type) to be a shortcoming.

The submitted manuscript lacks innovation, which can be followed up by a professional (scientific) society. The outputs are oriented towards a political-economic strategy with the aim of minimizing the production of emissions by diversifying the installation of new sources of air pollution and using the best available technologies.

The submitted article is at the required scientific level and I have no objections to it in the review process.

The article should be published after considering the comments in this review.

Author Response

Response to Reviewer 1 Comments

Point 1: Line 97 – why does α equals 0.5?

Response 1: Regarding line 97, why α is equal to 0.5, this is due to our reference to the conclusions of other literature (Qiu Fangdao et al., 2013).

Point 2: Table 1 – what does thy symbol *** mean in this table?

Response 2: In Table 1, the symbols *** in this table mean p < 0.01.

Response 3: The introduction has been modified in some way. A description of the structure of the paper is added to the introduction, and references are added and modified, etc.

Response 4: The research hypothesis is added, and the addition of relevant literature is completed.

Response 5: A description of the research methods and models used is provided.

Response 6: The purpose of the study is added to the abstract and some modifications are made to the research analysis.

Reviewer 2 Report

The authors of the article state that even though many companies did not operate during the Covid-19 pandemic, most of the cities in northern China suffered from pollution. The authors argue that this mainly occurred because of the heating demand in winter, and steel, thermal power, and petrochemical. The spillover effect occurred due to the excessive agglomeration of steel, thermal power, petrochemical, and other industries.

The article analyses the spatial spillover effect of the agglomeration of Thermal Power etc Air Pollution-Intensive Industries on environmental pollution and aims to facilitate the spatial planning and adjustment of Thermal Power etc Air Pollution-Intensive Industries in China.

In the end, the authors reach grounded and augmented conclusions and present valuable policy suggestions.

To conclude the article is interesting and conclusions valuable. However, I would suggest authors more precisely describe the methodology they use to reach the results. Just to insert the additional chapter and shortly to present the methodology which was used to reach the aims of the article. 

Author Response

Response to Reviewer 2 Comments

Dear reviewers, thank you for your care and conscientiousness in pointing out problems in the article to us. We have revised accordingly.

  1. The introduction has been modified in some way. A description of the structure of the paper is added to the introduction, and references are added and modified, etc.
  1. The structure of the article is revised and it is now divided into Introduction, Research Hypothesis, Research Methodology, Examination of Spatial Correlation and Setup of the Spatial Model, Analysis of the Results, Conclusions and Policy Suggestions.
  1. The research hypothesis is added, and the addition of relevant literature is completed.
  1. A description of the research methods and models used is provided.
  1. The purpose of the study is added to the abstract and some modifications are made to the research analysis.

Round 2

Reviewer 2 Report

The authors have duly taken into account the provided comments